# Energy Substrate Transporters in High-Grade Ovarian Cancer: Gene Expression and Clinical Implications

**DOI:** 10.3390/ijms23168968

**Published:** 2022-08-11

**Authors:** Marta Baczewska, Elżbieta Supruniuk, Klaudia Bojczuk, Paweł Guzik, Patrycja Milewska, Katarzyna Konończuk, Jakub Dobroch, Adrian Chabowski, Paweł Knapp

**Affiliations:** 1Department of Gynecology and Gynecological Oncology, Medical University of Bialystok, Marii Skłodowskiej-Curie 24A Street, 15-276 Bialystok, Poland; 2Department of Physiology, Medical University of Bialystok, Mickiewicza 2C Street, 15-222 Bialystok, Poland; 3Clinical Department of Gynecology and Obstetrics, City Hospital, Rycerska 4 Street, 35-241 Rzeszow, Poland; 4Biobank, Department of Medical Pathomorphology, Medical University of Bialystok, Waszyngtona 13 Street, 15-269 Bialystok, Poland; 5Department of Pediatric Oncology and Hematology, Medical University of Bialystok, Waszyngtona 17 Street, 15-274 Bialystok, Poland; 6University Oncology Center, University Clinical Hospital in Bialystok, Marii Skłodowskiej-Curie 24A Street, 15-276 Bialystok, Poland

**Keywords:** ovarian cancer, glucose transporters, fatty acid transporters, monocarboxylate transporter, amino acid transporters

## Abstract

Ovarian cancer is a non-homogenous malignancy. High-grade serous carcinoma (HGSC) is the most common subtype, and its drug resistance mechanisms remain unclear. Despite the advantages of modern pharmacotherapy, high-grade ovarian cancer is associated with a poor prognosis and research into targeted therapies is in progress. The aim of the study was to assess the dominant energy substrate transport mechanism in ovarian cancer cells and to verify whether genomic aberrations could predict clinical outcomes using the Cancer Genome Atlas (TCGA) dataset. Total RNA was extracted from HGSC frozen tissues, and the expression of selected genes was compared to respective controls. *GLUT1*, *FABPpm*, *MCT4* and *SNAT1* genes were significantly overexpressed in carcinomas compared with controls, while expression of *CD36/SR-B2*, *FATP1*, *FABP4*, *GLUT4*, *ASCT2* and *LPL* was decreased. No differences were found in *FATP4*, *LAT1*, *MCT1* and *FASN*. The transcript content of mitochondrial genes such as *PGC-1α*, *TFAM* and *COX4/1* was similar between groups, while the *β-HAD* level declined in ovarian cancer. Additionally, the *MCT4* level was reduced and *PGC-1**α* was elevated in cancer tissue from patients with ‘small’ primary tumor and omental invasion accompanied by ascites as compared to patients that exhibited greater tendencies to metastasize to lymph nodes with clear omentum. Based on TCGA, higher *FABP4* and *LPL* and lower *TFAM* expression indicated poorer overall survival in patients with ovarian cancer. In conclusion, the presented data show that there is no exclusive energy substrate in HGSC. However, this study indicates the advantage of glucose and lactate transport over fatty acids, thereby suggesting potential therapeutic intervention targets to impede ovarian cancer growth.

## 1. Introduction

Ovarian cancer (OC) is the seventh overall and third gynecological most common cancer in women and is still one of the major causes of mortality with a five-year overall survival rate below 45% in advanced stages [1,2]. Most patients are diagnosed at advanced stages due to non-specific symptoms at the early stage, low efficiency of the diagnostic methods and a lack of screening tests to detect the cancer [3]. Although the effects of chemotherapy in OC are initially satisfactory, most OC cases exert major therapeutic implications during treatment and pose the threat of cancer recurrence after first-line treatment [4].

OC is a non-homogenous malignancy which varies in genetic features, molecular morphology, clinical implications and therapy [5]. High-grade serous carcinoma (HGSC) is the most common subtype, accounting for over 70% of epithelial ovarian cancers (EOCs). The hallmark feature of HGSC is the diversity in the morphology of different cancer cells, depending on their intratumoral spatial localizations and the tumor microenvironment [6], leading to widely diverse drug resistance mechanisms [7]. 

The development of ovarian cancer requires an adequate energy supply, which is facilitated by the interactions between cancer cells and the tumor milieu [8,9]. The heterogeneous nature of OC manifests through the co-presence of cells highly dependent on glucose that shift their metabolism towards anaerobic processes even in the presence of oxygen, and ovarian cancer cells that rely on aerobic processes such as oxidative phosphorylation [10]. Glycolysis not only enables fast provision of ATP, but also generates intermediates that are further incorporated in the biosynthesis of amino acids and lipids. Free fatty acids play a crucial role in energy production, membrane synthesis in cancer cells and oncogenic signaling [11], while their bioavailability is enhanced in a neoplastic environment [12]. As part of metabolic alterations, elevated rates of fatty acid synthesis due to increased expression of various lipogenic enzymes could account for cancer progression [11]. Moreover, if phospholipids with saturated fatty acids are abundantly expressed in cancer membranes, cancer cells are less prone to respond to chemotherapy, which also highlights the role of free fatty acids in cancer physiology [13]. Amino acids are indispensable for tumor growth as they provide nitrogen for purine and pyrimidine synthesis, serve as precursors in protein and glutathione biosynthesis, and are involved in a vast number of tumor signaling and gene expression pathways, e.g., extracellular signal-regulated protein kinase (ERK) or mammalian target of rapamycin (mTOR) cascades [14,15]. In particular, an increase in glutamine dependency in high-invasive OC preserves mitochondrial integrity by supporting reactive oxygen species (ROS) scavenging mechanisms and replenishing tricarboxylic acid cycles for energy generation [16]. Differences in predominantly used energy substrates occur even in patients with similar OC histopathological type, underlining the importance of metabolic adjustments in cancer progression [17]. According to The Cancer Genome Atlas, HGSC can be classified into immunoreactive, proliferative, differentiated, and mesenchymal cell types based on RNA sequencing and microarray data analysis [18]. Fatty acid, glucose and amino acid transporters contribute to the development, growth and metastasizing of OC, and thus become potential targets for energy metabolism-based molecular and novel cancer therapy in OC patients.

A substantial correlation between metabolic disturbances occurring in obesity and the development of human cancers was reported in both animal models and clinical trials [19,20,21]. Obesity triggers a wide range of inflammatory effects, which induce a vast number of biological consequences, such as ROS and cytokine production and other mechanisms, by altering the insulin/IGF-1 axis or production of steroid hormone, hence promoting oncogenesis [22]. Although OC is not suspected to be an obesity-related cancer, some studies revealed a link between enhanced body mass index (BMI) and the likelihood of developing OC, although these findings were not confirmed by other studies [23,24,25]. It is known that the biochemical interplay between adipocytes and OC cells might implicate invasive properties in OC and mediate carboplatin resistance. Moreover, advanced or recurrent OCs have the propensity to metastasize to adipose-rich tissue such as the omentum [26].

A large body of evidence suggests that targeting energy substrate transport in poor prognostic patients with metabolically aggressive OC may be a plausible opportunity to enhance patients’ survival by introducing novel treatment schemes into conventional treatment. Therefore, the aim of this study was to assess metabolic and anthropometric factors in OC and their association with the level of expression of diverse energy substrate transporters in OC.

## 2. Results

### 2.1. Patient Characteristics

General characteristics of the control and study groups are presented in Table 1. The study group comprised 27 patients with HGSC, of which 23 patients were diagnosed with FIGO stage III–IV. Four patients carried the BRCA 1 or BRCA 2 tumor mutation. Interestingly, during surgical procedures (cytoreductive surgeries in HGSC) we observed two subgroups. Some patients presented a ‘small’ primary tumor and omental invasion accompanied by ascites (in our study group, seven patients). The other group had a greater tendency to metastasize to lymph nodes and omentum was clear (three patients). FIGO classification does not distinguish patients between these features. Based on in vivo research, OC cells have an affinity for fat tissue in the omentum. One of our hypotheses is that there might be molecular subgroups of HGSC that cause this heterogeneity (Table 2).

### 2.2. Energy Substrate Transporters and Metabolism-Related Gene Expression

Among the tested fatty acid carriers, gene expression of cluster of differentiation 36/a scavenger receptor class B protein (*CD36/SR-B2*) and fatty acid transport protein 1 (*FATP1*) was lower in OC compared to controls (−37% and −62%, respectively). The mRNA level of membrane associated fatty acid binding protein (*FABPpm*) was upregulated (+89%), while *FATP4* expression remained relatively constant between groups. The transcript content for cytosolic fatty acid binding proteins such as *FABP4* was diminished in OC relative to control samples (−93%; Figure 1a–e and Appendix A). There were no significant changes in *FASN* expression that would indicate enhanced de novo synthesis of fatty acids in cancer cells. The level of lipoprotein lipase (*LPL*), required for hydrolytic release of fatty acids from triacylglycerols in circulating lipoprotein particles, lowered by 78% in ovarian cancer (Appendix A).

The expression of glucose transport proteins was enhanced in the case of *GLUT1* (+18-fold), but decreased for *GLUT4* (−90%; Figure 1f,g and Appendix A). Regarding monocarboxylate transporters, increased transcript content was noticed for *MCT4* (+4-fold), but not for *MCT1* (Figure 1h,i and Appendix A). Importantly, the relative mRNA expression of *GLUT1* and *MCT4* in control ovarian samples was the lowest from all the tested energy substrate transporters, and increased to the highest extent during cancer development (Appendix A).

Furthermore, we demonstrated that the mRNA level of Na^+^-independent neutral amino acid transporter (*LAT1*) was comparable in both groups, while Na^+^-dependent neutral amino acid transporter expression declined in OC (*ASCT2*, −44%; Figure 1j,k and Appendix A). By contrast, the expression of Na^+^-coupled neutral amino acid transporter 1 (*SNAT1*) was markedly elevated in OC (+221%; Figure 1l and Appendix A).

Primary OC that developed metastasis was characterized by a lower expression of *LPL* (−59%) compared to tumors that had not spread to the lymph nodes. *MCT4* level was markedly diminished (−57%) in patients with ‘small’ primary tumor and omental invasion accompanied by ascites compared to cancer tissue from patients with greater tendency to metastasize to lymph nodes with clear omentum. Regarding other genes, we did not notice significant differences in their levels with respect to the FIGO stage as well as the presence of lymphatic node invasion or ‘omental-cake’ (Table A1).

### 2.3. Mitochondrial Gene Expression

In the next step, we verified the expression of several mitochondrial genes and observed a significant decline in *β-HAD* level in OC (−60%) compared to controls. We did not notice considerable differences in the transcript content for *PGC-1α*, *TFAM* and *COX4/1* in cancer tissue (Figure 2 and Appendix A). The expression of these genes did not differ with respect to the FIGO stage as well as the presence or absence of lymphatic node invasion or omentum ‘omental-cake’. Nonetheless, *PGC-1α* level was enhanced (+14-fold) in patients with ‘small’ primary tumor and omental invasion accompanied by ascites, compared to cancer tissue from patients characterized by a greater tendency to metastasize to lymph nodes with clear omentum (Table A1).

### 2.4. Associations of Gene Expression with BMI in Patients with Ovarian Cancer

Overweightness, defined as a BMI greater than 25 kg/m^2^, was not related to changes in transcript content in the studied genes. Obesity (BMI > 30 kg/m^2^), however, was associated with higher expression of *FABPpm, PGC-1α* and *FASN* in OC compared to tissue samples obtained from non-obese patients (Table 3).

### 2.5. Correlations

A correlation analysis among the values of expression of energy substrate transporters in control samples revealed negative associations between *FABP4* and *GLUT1* (*p* = 0.001, r = −0.807) as well as *LAT1* and *SNAT1* (*p* = 0.037, r = −0.569). Positive relationships involved *FABP4* with *GLUT4* (*p* = 0.02, r = 0.622), *FATP4* and *MCT1* (*p* = 0.0003, r = 0.846), *FATP4* and *MCT4* (*p* = 0.005, r = 0.723), *MCT1* and *GLUT1* (*p* = 0.013, r = 0.657), and *MCT4* and *ASCT2* (*p* = 0.03, r = 0.587). In cancer samples, we observed significant correlations between *FABP4* and *FABPpm* (*p* = 0.009, r = 0.490), *FATP4* and *MCT1* (*p* = 0.009, r = 0.489), *FATP4* and *FABPpm* (*p* = 0.0004, r = 0.636), and *SNAT1* and *GLUT1* (*p* = 0.005, r = 0.527) (Figure 3).

The positive associations that were present in both control and cancer samples included relationships between *FABP4* and *CD36/SR-B2* (control: *p* = 0.008, r = 0.692; cancer: *p* = 0.0003, r = 0.647), *FATP4* and *LAT1* (control: *p* = 0.037, r = 0.569; cancer: *p* = 0.012, r = 0.478), *FATP4* and *ASCT2* (control: *p* = 0.0001, r = 0.868; cancer: *p* = 0.03, r = 0.418), *MCT1* and *ASCT2* (control: *p* = 0.0003, r = 0.903; cancer: *p* = 0.01, r = 0.488) as well as *MCT4* and *LAT1* (control: *p* = 0.027, r = 0.596; cancer: *p* = 0.044, r = 0.398) (Figure 3).

Additionally, in control samples we noticed a negative relationship between *TFAM* and *FABP4* (*p* = 0.042, r = −0.556), *COX4/1* and *FATP4* (*p* = 0.011, r = −0.666), *COX4/1* and *MCT4* (*p* = 0.011, r = −0.789), as well as *FASN* and *FABP4* (*p* = 0.008, r = −0.692). Positive relationships were observed for *β-HAD* with *FATP1* (*p* = 0.038, r = 0.565), *MCT1* (*p* = 0.002, r = 0.767) and *ASCT2* (*p* = 0.01, r = 0.675), as well as for *TFAM* with *GLUT1* (*p* = 0.005, r = 0.719), *MCT1* (*p* = 0.044, r = 0.552) and *SNAT1* (*p* = 0.042, r = 0.556). In OC, there were positive relationships between *PGC-1α* and *CD36/SR-B2* (*p* = 0.035, r = 0.407), *FABPpm* (*p* = 0.005, r = 0.526), *FATP4* (*p* = 0.004, r = 0.536), *FABP4* (*p* = 0.006, r = 0.517) and *GLUT4* (*p* = 0.025, r = 0.431). Correlations common to control and OC were between *β-HAD* and *FATP4* (control: *p* = 0.011, r = 0.666; cancer: *p* = 0.003, r = 0.546) and *FASN* and *GLUT1* (control: *p* = 0.001, r = 0.824; cancer: *p* = 0.002, r = 0.563; Figure 3).

Moreover, positive correlations were found between the expression of *FABPpm* and BMI, *GLUT1* and plasma glucose concentration, and *LAT1* and tumor volume (Appendix A).

### 2.6. Genetic Alterations in Metabolism-Related Genes Based on TCGA and GTEx Datasets

In the next step, we verified whether our results correspond with data from The Cancer Genome Atlas (TCGA) and Genotype-Tissue Expression (GTEx) projects, which provide comprehensive transcriptomic data in a large number of patients with OC compared to normal ovarian tissue. We noticed only a few discrepancies between our data and TCGA data; namely, in a large OC cohort there were no alterations in *CD36/SR-B2*, *ASCT2* and *β-HAD* level (a decrease in our study) and *FABPpm* (an increase in our study). Additionally, while we did not notice alterations in *FASN* expression, in the TCGA-OC cohort this gene expression was elevated (Figure 4).

### 2.7. Prognostic Value of the Metabolic Pathway Genes Based on TCGA Cohort

To investigate the relationship between the expression of metabolism-related genes in OC and the clinical data, we analyzed overall survival, progression-free survival and clinical stage-related level for each gene. For overall survival analysis, higher *FABP4* and *LPL* and lower *TFAM* indicated poorer prognosis (Figure 5a and Appendix A). In the case of progression-free survival, higher expression of *FABP4*, *PGC-1α* and *COX4/1* indicated worse, while higher levels of *GLUT4* and *TFAM* correlated with better patient progression-free survival (Figure 5b and Appendix A). Furthermore, the expression of *FABPpm*, *FABP4*, *FASN*, *GLUT1* and *TFAM* correlated with the clinical stage of OC (Figure 6 and Appendix A).

## 3. Discussion

Metabolic reprogramming has been recognized as a hallmark of tumor development because it is required to sustain the energy supply for cancer progression and metastatic dissemination. The use of nutrients depends on their availability in the tumor microenvironment and reflects the capacity for cellular genetic alterations in response to stress conditions. To explore metabolite reliance in OC, we assessed the mRNA expression of numerous proteins required for transport of fatty acids (*CD36/SR-B2*, *FABPpm*, *FATP1*, *FATP4*), glucose (*GLUT1*, *GLUT4*), monocarboxylates (*MCT1*, *MCT4*) and amino acids (*LAT1*, *ASCT2, SNAT1*), and also evaluated the levels of several lipid metabolism-related (*FASN*, *LPL)* and mitochondrial genes (*PGC-1α*, *TFAM*, *β-HAD* and *COX4/1*). Moreover, association analysis of the above parameters with the clinical and histological characteristics of patients was performed. Finally, the transcriptome profiles from the TCGA-OC dataset were used to associate the above-mentioned genes’ expression with the clinical prognosis of OC patients.

### 3.1. Glucose Transporters

In our study, the *GLUT1* transcription level exerted a significantly higher expression in HGSC tissues compared to controls. This result is consistent with previous studies, which revealed that *GLUT1* mRNA and protein expression were enhanced in primary OC compared to other glucose transporters [27,28]. Moreover, concerning the stage of OC lesions, Cantuaria et al. revealed that there is a considerable correlation between the stage of OC and the expression of GLUT1, indicating that GLUT1 level is enhanced in malignant tumors compared to borderline tumors [29]. The hallmark feature of cancer cells is elevated glucose utilization by anaerobic processes such as glycolysis, even in the presence of oxygen, which is known as Warburg effect [30]. As a result of inefficient ATP production in anaerobic processes compared to oxidative phosphorylation, cancer cells are highly dependent on an enhanced influx of glucose [31]. In order to facilitate glucose uptake to maintain adequate energy supply, cancer cells enhance the expression of key factors of the glycolytic pathway, including glucose transporters (GLUTs) as well as other glycolytic enzymes [32]. The regulation of GLUT1 expression may be the reason why this glucose transporter is most prominently expressed in OC compared to other GLUTs. OC has the propensity to metastasize to the peritoneum; thus, OC cells and intraperitoneal metastases are susceptible to oxygen deprivation in the cancer microenvironment, and therefore hypoxia influences adaptive mechanisms in OC [33]. Hypoxia-inducible factor (HIF) is the main triggering factor responsible for the upregulation of GLUT1 [34]. Therefore, adaptive mechanisms to hypoxia may be a possible reason for the augmentation of GLUT1 expression in OC. In addition, genetic alterations such as P53 mutation, which is a triggering factor in developing OC, also contribute to GLUT1 upregulation [32]. On the other hand, the regulation mechanism of GLUT4 expression is not dependent on HIF-1α but rather is triggered by insulin. In our study, the expression of *GLUT4* was decreased and there is scarce information regarding the role and regulation of GLUT4 and insulin in OC [35]. Finally, the overexpression of GLUT1 is related to high-grade and poorly differentiated tumors in OC and is associated with a detrimental malignant nature of the cancer and a dismal prognosis [8,29]. Interestingly, the application of GLUT1 inhibitor BAY-876 in OC diminished the tumor growth by 50–71%, and decreased glucose influx and consumption in both in vitro and in vivo murine models [36]. Interestingly, ciglitazone, an antidiabetic drug, exerted an anti-proliferating effect on OC cells in vitro by altering GLUT1 abundance in the plasma membrane, thus regulating glucose utilization and mitigating the rapid growth of OC [37]. Other research revealed that the GLUT1 inhibitor STF 31 and metformin, combined with chemotherapy, increased therapy efficiency in sensitive and resistant OC [38]. In addition, targeting GLUT4 appears to have therapeutic potential to hinder the development of OC. A recent study reported that apatinib, which inhibits VEGFR2/AKT1/GSK3β/SOX5/GLUT4 pathway, decreased glucose utilization in animal models in OC and exerted an anti-proliferating effect [39].

### 3.2. Monocarboxylate Transporters

The enhanced expression of *MCT4* mRNA in OC observed in our study is consistent with previous data that outline the augmented expression of MCT4 under hypoxic conditions and the prognostic value of MCT4 in several human cancers, e.g., cervical cancer, osteosarcoma, prostate cancer, gliomas, bladder cancer [40,41,42]. MCT4 is an H^+^-coupled symporter that exports an excessive amount of lactate from cancer cells and is prominently expressed in metastatic tumors in a highly hypoxic environment [43]. Due to the Warburg effect, the neoplastic microenvironment is abundant in lactate and pyruvate [44]. Recently, it was revealed that MCT4 has a higher affinity for lactate compared to pyruvate and can export lactate even to a high lactate environment (MCT1 and MCT2 do not present this feature). These characteristics indicate that the expression of MCT4 takes precedence in metastatic cancers because their cells divide extensively and produce huge amounts of lactate, leading to increased acidity in the extracellular environment. Thus, the overexpression of MCT4 is an adaptive mechanism to prevent the deleterious effects of an acidic environment and hypoxia in a neoplastic environment [43]. Furthermore, recent studies demonstrated that MCT4 expression is triggered by hypoxia and mediated by HIF-1α, while MCT1 expression is not enhanced in this case [45]. Highly hypoxic areas are usually increased in large tumor lesions, along with inadequate blood flow and ascites in the tumor milieu [6]. Therefore, HIF-1α can be a plausible factor contributing to resistance to carboplatin via alterations in cancer cell metabolism by enhanced MCT4 expression [46]. The observation mentioned above highlights that the microenvironment of OC and its concomitant hypoxia contribute to the metabolic switch that results in cancer progression and treatment implications. Moreover, MCT4 expression and activity in OC are Influenced by chaperone CD147, and both are associated with dismal overall survival in several cancers [47]. The expression of CD147 was enhanced in most ovarian tumors induced by hypoxia, wherein CD147 served as an ancillary molecule for MCT-mediated lactate transport [48]. It has been shown that increased levels of MCT1, MCT4 and chaperone CD147 contribute to the development of EOC, and were correlated with high-grade tumors and ascites but not with histological type. However, only enhanced expression of MCT4, not of MCT1 or CD147, correlated with the likelihood of OC recurrence. Additionally, augmented expression of MCT4, MCT1 and chaperone CD147 was coupled with elevated levels of MDR1–multidrug resistance marker in OC. The expression of these molecules was detected in all metastatic lesions of OC after chemotherapy, so it can be stated that these molecule-mediated processes can lead to chemotherapy resistance [49]. Therefore, the MCT4-dependent mechanisms that contribute to EOC relapse need to be elucidated, because this transporter can potentially be used in targeted therapy in drug-resistant OC. The study conducted on mice with OC revealed that elevated MCT4 concentration yielded increased mouse mortality [50]. Interestingly, enhanced expression of MCT4 was observed in tumor stroma, especially in cancer-associated fibroblasts in breast cancer, whereas MCT1 was preferentially expressed in epithelial cancer cells and contributed to the influx of lactate to cancer cells. Thus, catabolic stromal cells transport lactate to epithelial cells, which have to fulfill high requirements for energy substances [51].

Recent studies demonstrate that synergy of the Warburg effect and reverse Warburg (lactate efflux) is thought to be a basis of cancer metabolism; however, data on the alterations in expression of metabolism-related transporters in OC are limited [17]. There is compelling evidence that enhanced GLUT1 expression and MCT4 expression are strongly associated with cancer progression, while the augmented expression of these transporters may be an adaptive mechanism to increased anaerobic glycolysis even under normoxia in order to maintain an adequate energy supply to an intensely proliferating cancer cell [52]. To sum up, GLUT1 mediates glucose influx and MCT4 induces lactic acid efflux, which are interdependent processes in cancer metabolism [53]. The enhanced glycolysis and maintenance of pH equilibrium in the case of intensive lactate production are hallmark abilities that contribute to progression from in situ to invasive cancer [54]. While blocking an individual MCT has been ineffective to restrain cancer growth (i.e., cancer cells escape death using glucose and glutamine as energy sources, or alternatively overexpress the other MCT isoform), combined inhibition is cytotoxic when paralleled by loss of mitochondrial NAD^+^ regenerating capacity due to suppression of glycolytic ATP production [55].

### 3.3. Fatty Acid Transporters

Despite the upregulation of glucose transporters and glucose-dependent metabolism in several human cancers, there is scarce data on fatty acid transporters in malignant cells. In a nutrient-deprived microenvironment (limited glucose availability) cancer cells’ metabolism shifts and fatty acid β-oxidation has a prominent role in maintaining an adequate energy supply for cancer cells [56]. Due to the evolving evidence for the cooperation between OC and adipocytes in the omentum, the alterations in lipid metabolism in HGSC are regarded as crucial factors in dissemination to the peritoneum and omentum [57]. In our study, however, we noticed reductions in the level of fatty acid transporters such as *CD36/SR-B2*, *FATP1* and *FABP4*, which were associated with lower *LPL* and mitochondrial *β-HAD* levels. The upregulation of glucose-related transporters coupled with a decrease in the expression of fatty acid transporters could be the cause of malignant metabolic alteration in cancers. There is compelling evidence that FABP4 chaperone protein and CD36/SR-B2 constitute pivotal regulatory factors of fatty acid transport in human cancer and are upregulated in several human cancers, e.g., breast cancer, colon cancer, cervical cancer [58,59,60]. FABP4 was observed in abundance in metastatic OC, which is discrepant with our study. Recent studies indicate that adipocytes and OC cells inevitably cooperate in the lipid supply and that the fatty acid transport between adipocytes and OC cells is mediated by FABP4 [61]. Congruently, the overexpression of FABP4 was associated with cancer-related adipocytes, and FABP4 mediated the transport of lipids from adipocytes to OC cells. Moreover, the overexpression of FABP4 increased the likelihood of formation of peritoneal metastases in OC [26]. In another study, the overexpression of FABP4 was associated with an adaptive mechanism and resulted in a decrease in lipid droplet formation coupled with ROS production. However, Nieman et al. did not demonstrate in their study enhanced expression of FABP4 in OC cells incubated without adipocytes, and the expression of FABP4 was enhanced in metastases compared to primary ovarian tumors [62]. The possible explanation for this discrepancy with our results is the fact that our study was conducted on OC at primary sites but not on metastases of OC from the peritoneum, which is surrounded by a lipid-abundant microenvironment. Indeed, some researchers reported that the suppression of FABP4 diminished the ability of an enhanced 5-hydroxymethylcytosine formation in DNA to invade the high-lipid cancer milieu, leading to decreased expression of genes involved in OC dissemination and enhanced susceptibility to chemotherapy both in vitro and in vivo. Thus, there is plausible evidence that targeting fatty acid transporters can be used as an alternative therapy in OC metastases [26]. Taken together, a prominent role of FABP4 is associated with metastases in the peritoneum, which develop in a microenvironment abundant in lipids that necessitates a linkage with the bioavailability of lipids in visceral fat.

CD36/SR-B2 is involved in binding and subsequently mediating the influx of long-chain fatty acids, oxidized lipids and phospholipids to cancer cells [63]. The findings of recent studies showed that cancers with enhanced expression of CD36/SR-B2, induced by a high-fat diet or by palmitic acid in mouse models of human oral cancer, have a proclivity for metastasizing [64]. Abundant presence of CD36/SR-B2 in OC was observed with concomitant adipocytes in the microenvironment. Previous research proved that OC cell lines co-cultured with human adipocytes have enhanced expression of CD36/SR-B2, which augments fatty acid influx to cancer cells. However, other FA transporters, such as FABPpm, FATP1 and FATP4, remained unaffected. Accordingly, CD36/SR-B2 is overexpressed in peritoneal metastases in OC compared to primary OC [65], whereas our study was mostly conducted on OC at primary sites and diminished expression of *CD36/SR-B2* was detected. Nevertheless, promising results were demonstrated in animal models with the use of anti-CD36/SR-B2 antibodies, which exerted growth-inhibitory effects and considerably limited the number of metastases [64,65].

Unlike other fatty acid transporters, *FABPpm* showed increased transcript content in OC tissue compared to the control group, with a positive correlation between the expression of *FABPpm* and BMI. It is hard to interpret these data since our study is one of the first to investigate the expression of *FABPpm* mRNA in OC. Therefore, future studies are necessary to determine FABPpm’s role in cancer progression.

### 3.4. Amino Acid Transporters

A vast number of studies have demonstrated that the expression of amino acid transporters LAT1 and ASCT2 is enhanced in tumor tissues, e.g., colorectal cancer, suggesting a crucial role of amino acid supply in fulfilling the high proliferation rate in cancer cells [66]. However, scarce information is available concerning the relationship of amino acid transporter expression with cell growth in human ovarian cancer. In our study, we showed that the level of *ASCT2* expression was decreased, while *LAT1* mRNA expression remained similar in both control and cancer groups. Previous studies demonstrated that LAT1 expression was increased in vitro in OC cell lines. However, the suppression of LAT1 did not cause a significant decrease in cell anchorage-dependent growth of both SKOV3 and IGROV1 cells in a liquid medium [67]. This finding suggests that the increase in LAT1 expression in the ovarian cell lines is not triggered by the OC cell proliferation mechanism, but rather that there are some other pathophysiological mechanisms that cause the increase in LAT1 expression in this cell line. These results also indicate that other energy substrate transporters play a more prominent role in OC cell proliferation, because the inhibition of LAT1 action does not alter the growth of OC cells. The lack of *LAT1* alterations in our study may be due to the fact that study [67] was conducted on OC cell lines, whereas ours was based on surgical specimens. Moreover, another study revealed that LAT1 shows enhanced expression in clear cell carcinoma and low presence in serous carcinoma compared to other histological types. This result is consistent with results of our study conducted on HGSC. These findings could prove that the overexpression of LAT1 is associated with certain OC types, especially clear cell carcinoma [68,69]. Furthermore, LAT1 expression was enhanced in G1 adenocarcinoma compared to G3 adenocarcinoma. These results are consistent with our studies, conducted mostly on advanced OC (FIGO III and IV in 23 out of 27 patients), which reported no differences in LAT1 mRNA expression. That study also proved the inverse correlation between LAT1 expression and p53 expression [70]. This phenomenon acknowledges that high LAT1 expression is not associated with the genetic basis of HGSC.

ASCT2 provides the compounds for tumor growth and progression but also maintains an adequate energy supply [71]. Literature results describing the relationships involving *ASCT2* mRNA expression in OC are inconsistent. Some studies reported that ASCT2 expression has a linkage with FIGO stage and a correlation with unfavorable overall survival in OC patients [72]. TCGA and other studies demonstrated no alterations in *ASCT2* mRNA expression in OC cell lines [67], while herein we observed lowered *ASCT2* levels in OC.

Unlike the above transporters, which are important for rapid amino acid exchange between the cell and its milieu to maintain amino acid homeostasis, SNAT1 is known to mediate net glutamine uptake to sustain glutaminolysis at a level that supports malignant hallmarks [73]. In line with that, we showed that primary ovarian tumors exhibited elevated *SNAT1* expression to fulfill their proliferative drive. SNAT1 level was associated with survival time and metastasis status, while silencing of the gene in human melanoma [74] and breast cancer cell lines [75] promoted senescence and diminished cell migration rate, thus indicating it as an important target in anticancer therapy. Indeed, recent in vitro studies with functional inhibition of SNAT1 [76] or glutamine deprivation [77] reduced cancer cell proliferation and migration.

### 3.5. Mitochondrial Genes

Mitochondrial adaptive mechanisms are centrally important for cancer cell survival in the face of environmental stresses, including hypoxia and chemotherapeutic drugs. The role of PGC-1α, a major regulator of mitochondrial biosynthesis and antioxidant activator, in OC remains controversial, since some studies reported a significant increase in PGC-1α and its target (i.e., TFAM) protein levels in OC samples relative to controls [78], which could mediate chemoresistance by reducing apoptosis [79,80]. The knockdown of PGC-1α or TFAM thereby increased sensitivity to cisplatin [80]. Contradicting these observations, other studies showed that HGSCs exhibited higher expression of PGC-1α/TFAM compared to clear cell carcinoma, which was related to better prognosis and chemosensitivity to initial platin and taxane therapy [81]. These data were also supported by in vitro experiments, wherein PGC-1α overexpression led to the apoptosis of OC cells mediated by a decreased Bcl-2/Bax ratio [82]; therefore, high expression of mitochondrial markers (TFAM and TIMM23) was considered beneficial in HGSC cell lines [83]. Although, in the TCGA cohort and in our study, there were no significant differences between *PGC-1α* expression in control and cancer tissue, this transcriptional co-activator level varied relative to BMI and cancer characteristics (Table A1 and Appendix A). Therapeutic implications of these associations are still unclear, but the involvement of mitochondrial proteins in the pathogenesis of HGSC and its chemosensitivity are evident.

The overt limitation of our study was the limited number of tested samples, which could impede adequate interpretation of the energy substrate transporter expression in OC. Another obstacle is the composition of tested tissue samples. In our research, ovarian tissue contained both epithelial and stromal cells. However, OC grows from the ovarian epithelium. Thus, analysis of the discrepancies in expressed genes in OC compared to control samples may pose a threat of misinterpretation because of an abundance of ovarian stromal cells, which may differ in the expression of diverse energy substrate transporters [8].

## 4. Materials and Methods

### 4.1. Study and Control Group

The present study conforms with the guidelines delineated in the Declaration of Helsinki and was approved by the Ethics Committee at the Medical University of Bialystok (permission number APK.002.221.2021). The samples were obtained by Biobank team at Medical University of Bialystok immediately after primary tumor removal. The biobanking serves as a unique entity dedicated to the collection of patients’ biological material according to the Standard Operating Procedures, which ensure the highest quality systematic biobanking, novel imaging techniques, and advanced molecular analysis for precise tumor diagnosis and therapy: The Polish MOBIT project. Scraps were precisely selected, cut into pieces that consisted of endothelial and stromal cells, snap-frozen in liquid nitrogen and thereafter stored in tanks with liquid nitrogen. In total we obtained 158 OC tissues during surgical procedures. Samples were screened for eligibility, but 119 patients met the exclusion criteria (other histological type of OC, comorbidities such as diabetes, L-thyroxine intake, hyperlipidemia, BMI > 35, other metabolic disorders). In 12 cases technical problems occurred, or the quality of the material was unsatisfactory. Ultimately, the study cohort included 27 patients with HGSC (cases were diagnosed from 2017 to 2021). The control group included healthy ovarian tissues obtained from non-oncological patients (14 met the inclusion criteria). Additionally, we analyzed clinical parameters obtained before surgery and patient questionnaires (diet, smoking history, family history, ECOG Performance Status Scale (WHO-Zubrod score), signs and symptoms). None of the patients received any treatment (chemotherapy, radiotherapy, or hormone therapy) before surgery.

### 4.2. Real-Time PCR Analysis

Total RNA was extracted using the NucleoSpin RNA Plus Kit with RNase-free DNase I treatment (Ambion, Thermo Fisher Scientific, Waltham, MA, USA), according to the manufacturer’s protocol. RNA quantity and quality measurements were performed using spectrophotometry (at an absorbance OD ratio of 260/280 and 260/230). Total RNA (1 µg) served as a template for first-strand cDNA synthesis using the EvoScript universal cDNA master kit (Roche Molecular Systems, Boston, MA, USA). Quantitative real-time polymerase chain reaction (qRT-PCR) was performed using the LightCycler 96 System real-time thermal cycler with FastStart essential DNA green master (Roche Molecular Systems). The following reaction parameters were applied: 15 s denaturation at 94 °C, 30 s annealing at 60 °C for *CD36/SR-B2*, *FATP1*, *FATP4*, *FABPpm*, *FABP4*, *GLUT1*, *GLUT4, FASN* and *β-actin* or 61 °C for *MCT1*, *MCT4*, *LAT1*, *ASCT2*, *SNAT1*, *PGC-1α*, *TFAM*, *β-HAD*, *COX4/1* and *LPL*, followed by 30 s extension at 72 °C for 45 cycles. PCR product specificity was verified by melting curve analysis. Reactions were run in duplicates and the expression was normalized against the housekeeper gene (β-actin). Results were calculated using the relative quantification method modified by Pfaffl [84]. The primers used in the study are listed in Table 4.

### 4.3. Public Data Mining

To screen for transcriptional metabolic dysregulation in ovarian cancer from a large patient cohort, we used TCGA RNA-sequencing data from The Gene Expression Profiling Interactive Analysis (GEPIA) database. GEPIA contains validated gene expression data from the TCGA-OC cohort and the Genotype-Tissue Expression project (GTEx) based on normal ovarian tissues [85]. The association between the mRNA expression of metabolism-related genes in TCGA and overall survival and progression-free survival in ovarian cancer patients was evaluated using Kaplan–Meier plots [86].

### 4.4. Statistical Analysis

Statistical analyses were performed with GraphPad Prism software version 8.0 (GraphPad Software, Inc., San Diego, CA, USA). The Shapiro–Wilk test (test for normality) and Levene tests (test for homogeneity of variances) were used to determine the application of parametric or non-parametric methods. Afterwards, Student’s *t*-test or Mann–Whitney U test was used to compare the differences between the groups. For multiple comparisons, the Kruskal–Wallis test followed by Dunn’s post-hoc test was applied. The relationships between the analyzed parameters were assayed with Spearman’s correlation coefficient. A log-rank test was applied to compare high and low levels of metabolism-related gene expression in Kaplan–Meier curves. Statistical hypotheses were verified at the 0.05 significance level.

## 5. Conclusions

The successful outgrowth and aggressiveness of OC relies on cellular metabolic flexibility as a response to the microenvironmental context and nutrient availability (Figure 7). Our results suggest that the metabolism of ovarian cancer is highly reliant on glucose influx mediated by GLUT1. Moreover, given the fact that the OC environment is acidic [87], the concomitant increase in MCT4 plays a pivotal role in maintaining Ph balance. Glycolysis also provides indirect metabolic precursors for the biosynthesis of nonglucidic acids crucial in amino acid and lipid transformation [56]; therefore, it may be a principal metabolic process in cancer cells. At the same time, the upregulation of amino acid transporter *SNAT1* suggests high glutamine dependence in ovarian cancer cells. Overall, our results and TCGA data analysis reveal differences in the metabolic biology of healthy and cancer cells that indicate potential therapeutic intervention targets to impede ovarian cancer cells’ proliferation and metastasis.

## Figures and Tables

**Figure 1 ijms-23-08968-f001:**
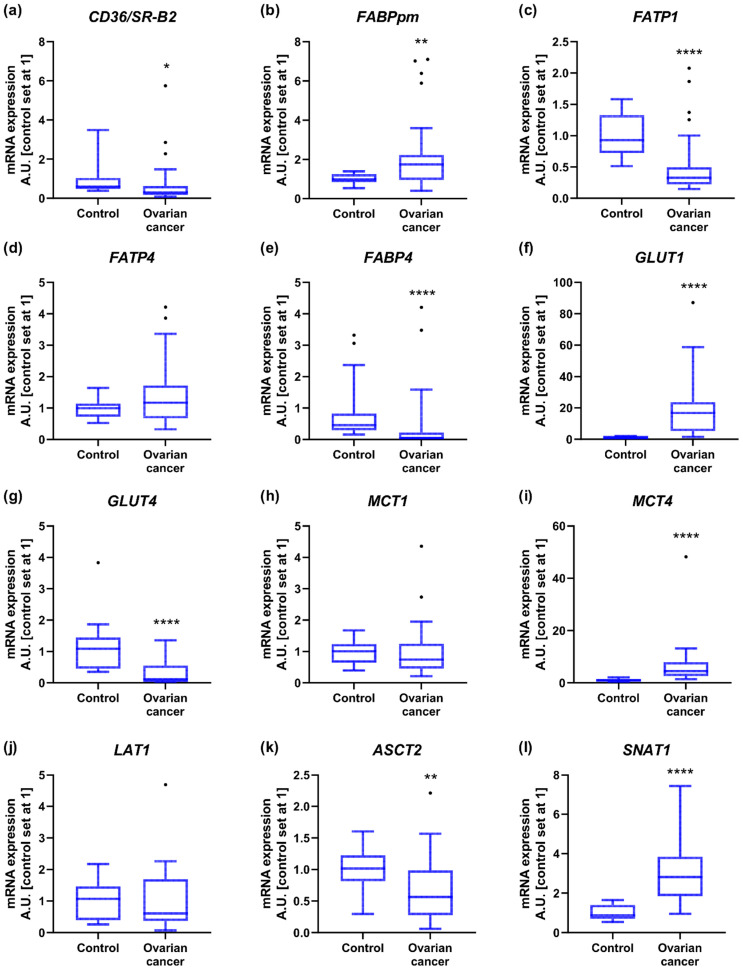
Transcript levels of (**a**) cluster of differentiation 36/a scavenger receptor class B protein, (**b**) membrane associated fatty acid binding protein, (**c**) fatty acid transport protein 1, (**d**) fatty acid transport protein 4, (**e**) fatty acid binding protein 4, (**f**) glucose transporter 1, (**g**) glucose transporter 4, (**h**) monocarboxylate transporter 1, (**i**) monocarboxylate transporter 4, (**j**) Na^+^-independent neutral amino acid transporter, (**k**) Na^+^-dependent neutral amino acid transporter and (**l**) Na^+^-coupled neutral amino acid transporter 1 in ovarian control (n = 14) and cancer tissue (n = 27). Measurements were made in duplicate, and arithmetic means were used for subsequent investigation. Results are expressed in arbitrary units with control set as 1 and presented as median (min to max) value. Differences statistically significant at: * *p* < 0.05, ** *p* < 0.01, **** *p* < 0.0001.

**Figure 2 ijms-23-08968-f002:**
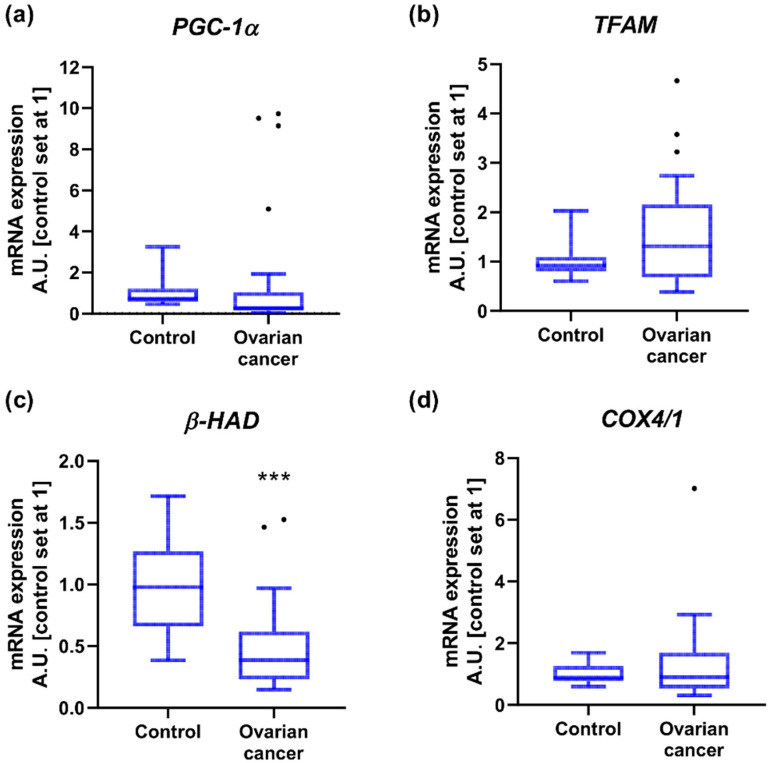
Transcript levels of (**a**) peroxisome proliferator-activated receptor gamma co-activator 1α, (**b**) mitochondrial transcription factor A, (**c**) acetyl-CoA acyltransferase and (**d**) cytochrome c oxidase subunit 4 isoform 1 in ovarian control (n = 14) and cancer tissue (n = 27). Measurements were made in duplicate and arithmetic means were used for subsequent investigation. Results are expressed in arbitrary units with control set as 1 and presented as median (min to max) value. Differences statistically significant at *** *p* < 0.001.

**Figure 3 ijms-23-08968-f003:**
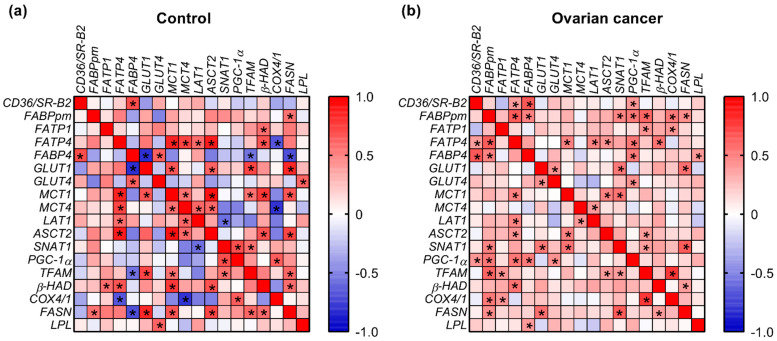
Heat map of correlations between the expression of energy substrate transporters and mitochondrial genes in (**a**) ovarian control and (**b**) cancer tissue. The relationships between the analyzed parameters were assayed via Spearman correlation coefficient. Correlations that were statistically significant (*p* < 0.05) are indicated with an asterisk.

**Figure 4 ijms-23-08968-f004:**
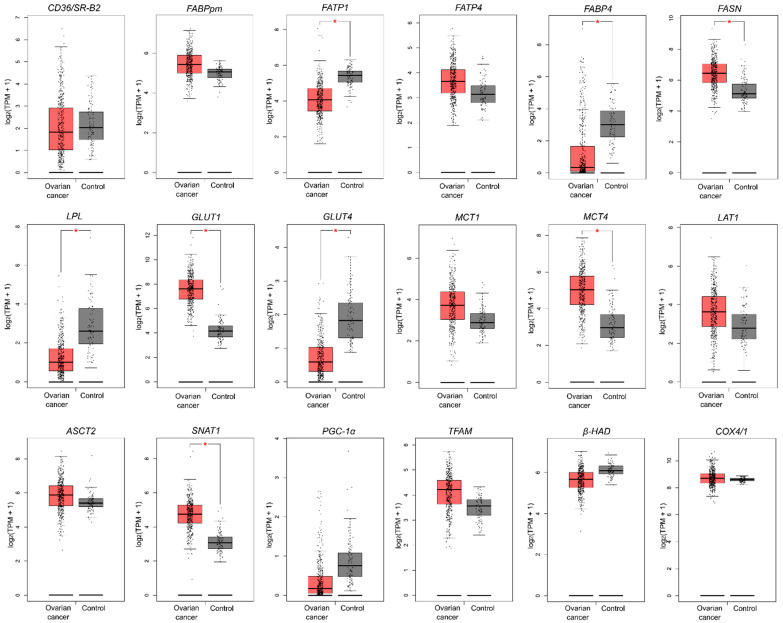
The transcription level of metabolism-related genes that significantly differed between ovarian cancer (n = 426) and control tissue (n = 88). Data are based on the TCGA cancer cohort and GTEx control samples. Data derive from The Gene Expression Profiling Interactive Analysis (GEPIA) database based on the TCGA-OC dataset and normal ovarian tissues (GTEx, Genotype-Tissue Expression project). Differences statistically significant at * *p* < 0.05.

**Figure 5 ijms-23-08968-f005:**
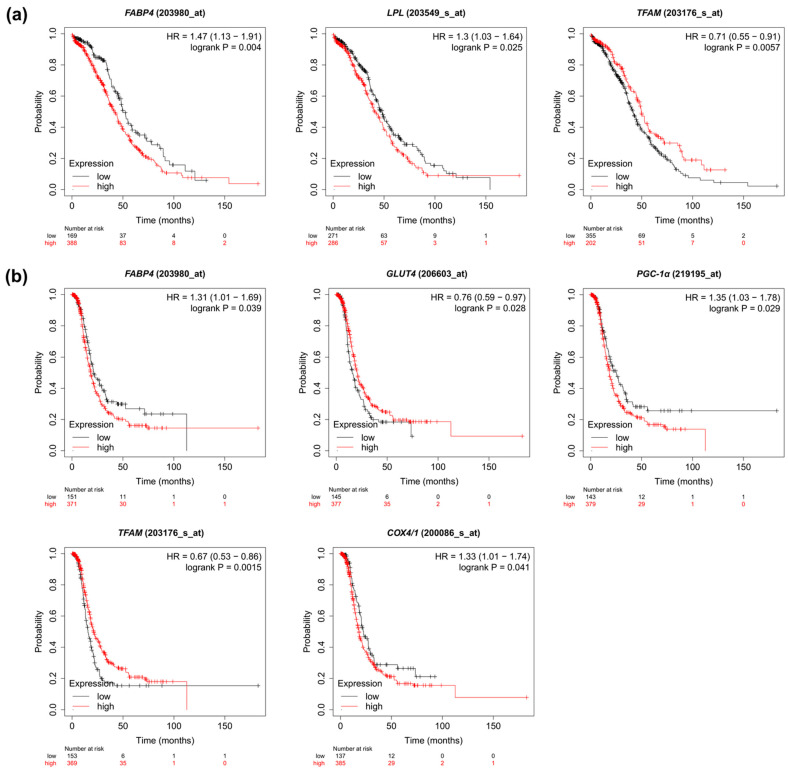
Kaplan–Meier curve analysis of (**a**) overall and (**b**) progression-free survival comparing high and low levels of the metabolism-associated genes.

**Figure 6 ijms-23-08968-f006:**
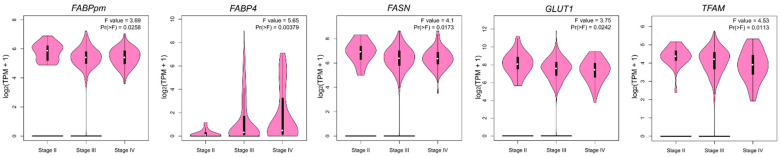
Violin plots representing significant correlations (*p* < 0.05) between gene expression and clinical stage based on the TCGA-OC dataset.

**Figure 7 ijms-23-08968-f007:**
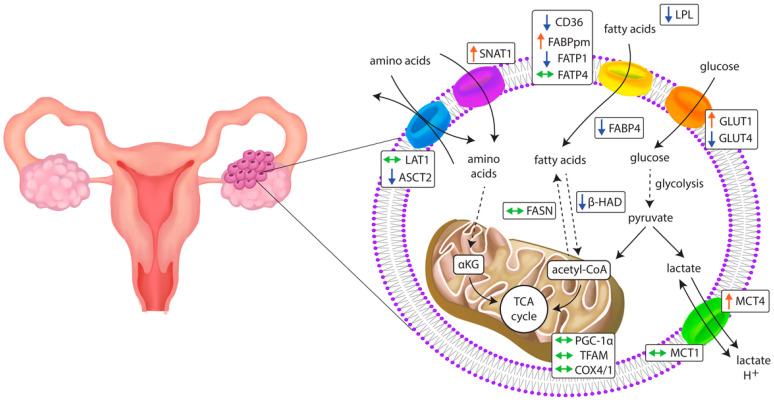
Changes in the expression of genes related to metabolic pathways in primary ovarian cancer tissue. ↑: increase; ↓: decrease; ↔: unchanged.

**Table 1 ijms-23-08968-t001:** Study and control group characteristics. Values are presented as median and interquartile range.

	Control	Ovarian Cancer	*p*
Total Age mean	n = 14 55.72 (45.58–63.75)	n = 27 63.56 (57.31–70.86)	- 0.06
BMI (kg/m^2^) Overweight/obese Ca125 (U/mL) PLT (x103 cells/mm^3^) Fibrinogen (mg/dl) Serum potassium (mEq/L) TSH (µU/mL) SBP (mmHg) DBP (mmHg) Primary tumor velocity ^1^ Time of hospitalization (day)	26.67 (24.92–28.72) n = 9 16 (10.6–26.0) 219 (206–260) 317 (288–356) 4.06 (4.0–4.3) 1.33 (1.18–1.6) 131 (124–149) 87 (83–90) - 4.0 (3.0–5.0)	27.89 (24.85–33.53) n = 19 503.00 (267.00–1478.00) 350.00 (266.0–452.0) 453.0 (373.0–522.0) 4.72 (4.39–5.10) 1.79 (1.32–2.43) 132 (130–145) 86 (73–92) 109.9 (64.11–276.32) 9.5 (7.0–14.0)	0.32 - <0.00001 0.00023 0.0014 0.0016 0.23 0.40 0.65 - 0.000024

^1^ Calculated using the formula π/6 × length × width × height. Abbreviations: BMI, body mass index; Ca125, cancer antigen 125; DBP, diastolic blood pressure; PLT, platelet count; SBP, systolic blood pressure; TSH, thyroid stimulating hormone.

**Table 2 ijms-23-08968-t002:** Histopathological characteristics of study group.

	n
Total	27
FIGO I	2
FIGO II	2
FIGO III	20
FIGO IV	3
BRCA 1/2 mutation	4
p53	14/19 ^1^
Wilms tumor gene product (WT1)	13/15
p16	1/2
Vimentin	0/6
Estrogen receptors (ERs)	5/12
Progesterone receptors (PRs)	2/5
Nodal invasion	15/27
Omentum ‘omental-cake’ ^2^	12/27
Nodal invasion > omental invasion ^3^	3
Nodal invasion < omental invasion ^4^	7
Cancer cells in peritoneal fluid	14

^1^ x/y; x: number of positive samples, y: number of checked samples, if not all from study group. ^2^ ‘Omental cake’ is a specific term used to describe this serious peritoneal disease with a mass-like feature ^3^ Number of lymph nodes involved >50% and omentum clear ^4^ Number of lymph nodes involved <50% and ‘omental cake’

**Table 3 ijms-23-08968-t003:** Log2-fold changes in gene expression in ovarian cancer tissue obtained from overweight (n = 9) or obese patients (n = 10) compared to lean individuals (n = 8).

Gene	Fold Change		*p* Value	
	Overweight	Obese	Overweight	Obese
*CD36/SR-B2*	0.9172	−0.1781	0.252	>0.999
*FABPpm*	0.6058	1.0061	0.329	**0.026**
*FATP1*	0.0376	0.552	>0.999	>0.999
*FATP4*	0.3567	0.8573	0.974	0.219
*FABP4*	2.1578	0.7444	0.51	0.288
*GLUT1*	−1.6832	0.3236	0.407	>0.999
*GLUT4*	0.737	2.214	0.779	0.094
*MCT1*	−0.7112	−0.3096	0.908	>0.999
*MCT4*	0.4707	−0.245	>0.999	0.622
*LAT1*	1.131	0.1321	0.856	>0.999
*ASCT2*	−0.2871	0.6762	0.827	0.75
*SNAT1*	−0.073	0.7698	>0.999	0.208
*PGC-1α*	1.1636	1.8047	0.089	**0.016**
*TFAM*	0.6391	1.0391	0.955	0.984
*β-HAD*	−0.575	0.2069	0.861	0.994
*COX4/1*	0.221	0.6918	>0.999	0.532
*FASN*	−0.2943	1.2679	0.948	**0.045**
*LPL*	0.5653	0.7144	0.888	0.201

**Table 4 ijms-23-08968-t004:** Primer sequences used for real-time PCR.

Target Gene	Forward Primer (5′-3′)	Reverse Primer (5′-3′)	Amplicon Length [bp]
*CD36/SR-B2*	GGTACAGATGCAGCCTCATT	AGGCCTTGGATGGAGAACA	157
*FATP1/SLC27A1*	GCTAAGGCCCTGATCTTTGG	CCAAGTCTCCAGAGCAGAAC	316
*FATP4/SLC27A4*	TGGCGCTTCATCCGGGTCTT	CGAACGGTAGAGGCAAACAA	140
*FABPpm*	GAAGGCAAAGGTGCGACAGT	GCCGAACGGTAGAGGCAAA	71
*FABP4*	GGGCCAGGAATTTGACGAAG	AACTCTCGTGGAAGTGACGC	184
*GLUT1/SLC2A1*	CACCACCTCACTCCTGTTAC	CCACTTACTTCTGTCTCACTCC	123
*GLUT4/SLC2A4*	GACCAACTAAGGCAAAGAG	CAATAGGATGCTTGTCTTCA	183
*MCT1/SLC16A1*	CACCGTACAGCAACTATACG	CAATGGTCGCCTCTTGTAGA	115
*MCT4/SLC16A3*	ATTGGCCTGGTGCTGCTGATG	CGAGTCTGCAGGAGGCTTGTG	243
*LAT1/SLC7A5*	CACAGAAAGCCTGAGCTTGA	CACCTGCATGAGCTTCTGA	249
*ASCT2/SLC1A5*	AGCTGCTTATCCGCTTCTTCAA	AGCAGGCAGCACAGAATGTA	175
*SNAT1/SLC38A1*	GCTTTGGTTAAAGAGCGGG	CTGAGGGTCACGAATCGGA	151
*PGC-1α*	AGCCTCTTTGCCCAGATCTT	GGCAATCCGTCTTCATCCAC	241
*TFAM*	AGCTCAGAACCCAGATGC	CCACTCCGCCCTATAAGC	115
*β-HAD*	CTTGCTCCGAGAGGGAGTC	AGCTCGTAGCTGGGAGGAAC	148
*COX 4/1*	GGTCACGCCGATCCATATAAG	TCTGTGTGTGTACGAGCTCATGA	79
*FASN*	CTTCCGAGATTCCATCCTACGC	TGGCAGTCAGGCTCACAAACG	131
*LPL*	GAGATTTCTCTGTATGGCACC	CTGCAAATGAGACACTTTCTC	276
*β-actin*	AGTCGGTTGGAGCGAGCATC	GGACTTCCTGTAACAACGCATCTC	115

## Data Availability

The data presented in this study are available on request from the corresponding author.

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
