# Peer review of "Energy Substrate Transporters in High-Grade Ovarian Cancer: Gene Expression and Clinical Implications"

_ijms, 2022, doi:10.3390/ijms23168968_

Round 1

Reviewer 1 Report

The authors of this paper measured the expression of energy substrate transporters and mitochondrial genes in ovarian control vs. cancer tissue and found the difference in expression (over or under) across multiple genes, mostly consistent with previous studies. However, this study, as it is, lacks the novelty and significance of clinical findings. Figures 1 and 2 and the remaining tables are redundant and should be included in the supplementary materials as this data has been summarized in Figure-3. 

Author Response

Dear Reviewer,

Thank you for giving us the opportunity to submit a revised version of the manuscript titled ‘Energy Substrate Transporters In High-Grade Ovarian Cancer - Gene Expression And Clinical Implications’. We appreciate the time and effort that You have dedicated to providing Your valuable feedback on the manuscript. For Your greater convenience, we have highlighted the changes within the manuscript using red font. We have put the responses to your comments below:

Point 1: The authors of this paper measured the expression of energy substrate transporters and mitochondrial genes in ovarian control vs. cancer tissue and found the difference in expression (over or under) across multiple genes, mostly consistent with previous studies. However, this study, as it is, lacks the novelty and significance of clinical findings.

Response 1: Thank you for pointing this out. We agree with this comment, so that to improve the manuscript we compared or results with data from a larger study cohort, i.e.,  The Cancer Genome Atlas (TCGA) and Genotype-Tissue Expression (GTEx) project that provide comprehensive transcriptomic data in a large number of patients with ovarian cancer and corresponding normal ovarian tissue. We assessed the predictive value of analyzed metabolism-related genes with overall and progression-free survival in OC patients. In the discussion section, we also added the potential therapeutical approaches aimed at targeting metabolism-related genes, although they are still in the initial phases.

We hope You will find these changes in the manuscript satisfactory.

Point 2: Figures 1 and 2 and the remaining tables are redundant and should be included in the supplementary materials as this data has been summarized in Figure-3.

Response 2: Thank you for pointing this out. However, we cannot fully agree that Figure 3 summarizes data from both the Figures 1 and 2. We consulted this with a statistician and it turns out that the median/mean in a group does not necessarily means correlation between them. As an example let's consider two groups of data.

A   B

1   3

2   6

3   9

4   12

Mean: 2.5 and 7.5; correlation coefficient: 1

Now, when we shuffle the data in group B we will have:

A   B

1   12

2   3

3   6

4   9

the means did not change (2.5, 7.5), but the correlation is now: -0.2

We admit that previously Figure 3 was confounding. Now, for better readability and clarity, we have modified it. The heatmap of gene correlations was prepared with GraphPad Prism software version 8.0 (GraphPad Software, Inc., San Diego, CA) and correlations in control and cancer groups are presented separately. The relationships between the analysed parameters were assayed with Spearman correlation coefficient. Additionally, we indicated statistically significant correlations (p<0.05) with asterisks.

Once again, thank You for all the comments and suggestions on our manuscript. We hope that You will accept the incorporated modifications of the manuscript.

Best regards,

Marta Baczewska - corresponding Author

Reviewer 2 Report

Baczewska et. al. submitted their original research article entitled “Energy substrate transporters in high-grade ovarian cancer gene expression and clinical implications” to “The International Journal of Molecular Sciences”.

The writing and the information supplied in the manuscript are nicely conducted. The findings reported in the article are complementary to the review article the authors published in “The Journal of Molecules” in 2021. On the other hand, certain aspects need to be addressed by the authors and I suggest minor revisions for the manuscript to be considered for publication in “The International Journal of Molecular Sciences” which I listed below:

Comment #1: In the introduction section, the authors mentioned that ovarian cancer is the seventh most common cancer in women, but the reference is outdated. The authors should update the cancer census.

Sung, H.; Ferlay, J.; Siegel, R.L.; Laversanne, M.; Soerjomataram, I.; Jemal, A.; Bray, F. Global Cancer Statistics 2020: GLOBOCAN Estimates of Incidence and Mortality Worldwide for 36 Cancers in 185 Countries. CA A Cancer J. Clin. 2021, 71, 209–249.

Huang, J.; Chan, W.C.; Ngai, C.H.; Lok, V.; Zhang, L.; Lucero-Prisno, D.E., III; Xu, W.; Zheng, Z.-J.; Elcarte, E.; Withers, M.; et al. Worldwide Burden, Risk Factors, and Temporal Trends of Ovarian Cancer: A Global Study. Cancers 2022, 14, 2230. https:// doi.org/10.3390/cancers14092230

Comment #2: In the introduction section, the authors mentioned most about ovarian cancer and less about the metabolic pathways that drive cancer development. The background regarding ovarian cancer should be shortened, and the information about the cancer-related energy metabolism should be expanded.

Comment #3: The authors can include TCGA data to show whether the metabolic pathway genes are associated with differential overall and progression-free survival in patients with ovarian cancer. In this way, they can integrate the clinical data of a large number of patients with the experimental findings of this article.

Comment #4: In section 2.5. Correlations, the figure they refer to should be Figure 3, not Figure 2.

Comment #5: The method that was used for the heatmap of gene correlations is missing and should be included in the manuscript.

Comment #6: As the authors mentioned in the ‘limitations’ section, the tumor samples obtained from ovarian cancer patients are composed of not only tumor cells but also stromal cells. Therefore, I suggest including an imaging technique (e.g., immunohistochemistry) to prove that the tumor cells are indeed the source of aberrantly expressed metabolic pathway genes.  

Comment #7: The authors should elaborate on potential therapeutical approaches to target the molecules associated with pro-tumorigenic metabolic pathways in ovarian cancer patients.

Comment #8: A summary figure explaining the aberrantly expressed genes and their effect on ovarian cancer progression would be informative for the readers at first glance.

Author Response

Dear Reviewer,

Thank You very much for the in-depth analysis of our study. We appreciate the time and effort You have dedicated to provide insightful comments on our paper. Below You will find our answers written in a point by point manner. Below You will find our answers to the discussed issues. For Your greater convenience we have placed them (as well as all the changes in the manuscript) with red font.

Point 1: In the introduction section, the authors mentioned that ovarian cancer is the seventh most common cancer in women, but the reference is outdated. The authors should update the cancer census.

Sung, H.; Ferlay, J.; Siegel, R.L.; Laversanne, M.; Soerjomataram, I.; Jemal, A.; Bray, F. Global Cancer Statistics 2020: GLOBOCAN Estimates of Incidence and Mortality Worldwide for 36 Cancers in 185 Countries. CA A Cancer J. Clin. 2021, 71, 209–249.

Huang, J.; Chan, W.C.; Ngai, C.H.; Lok, V.; Zhang, L.; Lucero-Prisno, D.E., III; Xu, W.; Zheng, Z.-J.; Elcarte, E.; Withers, M.; et al. Worldwide Burden, Risk Factors, and Temporal Trends of Ovarian Cancer: A Global Study. Cancers 2022, 14, 2230. https:// doi.org/10.3390/cancers14092230

Response 1: Thank you for pointing this out. We have updated the data and reference list according to Your suggestion.

Point 2: In the introduction section, the authors mentioned most about ovarian cancer and less about the metabolic pathways that drive cancer development. The background regarding ovarian cancer should be shortened, and the information about the cancer-related energy metabolism should be expanded.

Response 2: Thank you for this remark. According to Your suggestion, we have modified the Introduction. We shortened the background regarding ovarian cancer and included more detailed data regarding metabolism reprogramming that drives ovarian cancer development.

Point 3: The authors can include TCGA data to show whether the metabolic pathway genes are associated with differential overall and progression-free survival in patients with ovarian cancer. In this way, they can integrate the clinical data of a large number of patients with the experimental findings of this article.

Response 3: Thank you for pointing this out. To screen for transcriptional metabolic dysregulation in ovarian cancer from a large patient cohort, we used TCGA RNA-sequencing data using The Gene Expression Profiling Interactive Analysis (GEPIA) database. GEPIA contains validated gene expression data from TCGA-OC cohort and from Genotype-Tissue Expression project (GTEx) based on normal ovarian tissues. The association between the mRNA expression of metabolism-related genes in TCGA with overall survival and progression-free survival in ovarian cancer patients was evaluated using Kaplan-Meier Plotter (Materials and Methods, section 4.4). Most of the genomic alterations were similar between our study group and TCGA cohort. We noticed only few discrepancies between ours and TCGA data, namely in a large OC cohort there were no alterations in CD36/SR-B2, ASCT2 and β-HAD level (a decrease in our study) and FABPpm (an increase in our study). Additionally, while we did not notice alterations in FASN expression, in TCGA-OC cohort this gene expression was elevated (Section 2.6, Figure 4). To investigate the relationship between the expression of metabolism-related genes in OC with the clinical data, we analyzed overall survival, progression-free survival and clinical stage-related level of each gene (Section 2.7). For overall survival analysis, higher FABP4 and LPL, and lower TFAM indicated poorer prognosis (Figure 5a, Figure S3). In the case of progression-free survival, higher expression of FABP4, PGC-1α and COX4/1 indicated poorer, while higher level of GLUT4 and TFAM was related with better patient progression-free survival (Figure 5b, Figure S4). Furthermore, the expression of FABPpm, FABP4, FASN, GLUT1 and TFAM correlated with the clinical stage of OC (Figure 6, Figure S5). We hope that the inclusion of this data fulfils Your requirements.

Point 4: In section 2.5. Correlations, the figure they refer to should be Figure 3, not Figure 2.

Response 4: We are sorry for this mistake, we have corrected the Figure number.

Point 5: The method that was used for the heatmap of gene correlations is missing and should be included in the manuscript.

Response 5: Thank you for pointing this out. For better readability and clarity, we have modified Figure 3 and now, the correlations in control and cancer groups are presented separately. The heatmap of gene correlations was prepared with GraphPad Prism software version 8.0 (GraphPad Software, Inc., San Diego, CA). The relationships between the analysed parameters were assayed with Spearman correlation coefficient. Additionally, we indicated statistically significant correlations (p<0.05) with asterisks. This information was included in Figure description.

Point 6: As the authors mentioned in the ‘limitations’ section, the tumor samples obtained from ovarian cancer patients are composed of not only tumor cells but also stromal cells. Therefore, I suggest including an imaging technique (e.g., immunohistochemistry) to prove that the tumor cells are indeed the source of aberrantly expressed metabolic pathway genes. 

Response 6: Thank You for bringing our attention to this shortcoming of the manuscript. We would really like to include this data in the manuscript, however, to obtain the histological images we would require more time due to technical issues and holiday season in our team. We are sorry for these problems.

Point 7: The authors should elaborate on potential therapeutical approaches to target the molecules associated with pro-tumorigenic metabolic pathways in ovarian cancer patients.

Response 7: Thank you for raising this concern. Our results and previous preclinical studies indicate potential therapeutic intervention targets to impede ovarian cancer cells proliferation and metastasis. The co-administration of molecules targeting pro-tumorigenic metabolic pathways in ovarian cancer patients with conventional therapy seems to alleviate potential implications associated with chemotherapy resistance. Recent studies reported that the use of molecules influencing ovarian cancer metabolic pathways enhances the efficacy of the therapy and mitigates peritoneal metastasis in ovarian cancer. Preclinical studies indicate that influencing glycolytic pathway has a potential clinical application in OC treatment. Moreover, the inhibition of GLUT1 with inhibitor BAY-876 in OC in vitro and in vivo in mice at-tenuated tumor proliferation and  suppressed glucose utilization (Ma et al., 2019). Interestingly, ciglitazone, antidiabetic drug, exerted anti-proliferating effect on OC cells in vitro by altering GLUT1 abundance in the plasma membrane, thus regulated glucose utilization and mitigated the rapid growth of OC [92]. Other research revealed that STF 31- GLUT1 inhibitor and metformin combined with chemotherapy increased therapy efficiency in sensitive and resistant OC (Xintaropoulou et al, 2018). Also, targeting GLUT4 suggests therapeutic potential to hinder the development of OC. Recent study reported that apatinib, which inhibits VEGFR2/AKT1/GSK3β/SOX5/GLUT4 pathway, decreased the glucose utilization in animal models in OC and exerted anti-proliferating effect (Chen et al., 2019). Among other therapeutic targets are MCTs. While blocking of individual MCT has been ineffective (i.e., cancer cells escape death using glucose and glutamine as energy source, or alternatively overexpress the other MCT isoform), combined inhibition is cytotoxic when paralleled by loss of mitochondrial NAD+ regenerating capacity due to the suppressed glycolytic ATP production (Benjamin et al., 2018). Fatty acid transporters were also a subject of anti-cancer approaches. In mice xenograft models, the use of CD36 monoclonal antibodies reduced ovarian tumor weight and limit the number of metastasis (Ladanyi et al., 2018). Promising effects were also observed in murine oral cancer, where anti-CD36 antibodies caused almost complete inhibition of metastasis (Pascual et al., 2017). Targeting FABP4 diminished the ability to invade the high lipid cancer milieu by an enhanced 5-hydroxymethylcytosine formation in DNA leading to decreased expression of genes involved in OC dissemination and enhanced the susceptibility to chemotherapy both in vitro and in vivo (Mukherjee et al., 2020). Additionally, recent in vitro studies also showed that functional inhibition of SNAT1 (Böhme-Schäfer et al., 2022) or glutamine deprivation (Gwangwa et al., 2019) reduced cancer cell proliferation and migration.

These explanations were added to the manuscript.

Point 8: A summary figure explaining the aberrantly expressed genes and their effect on ovarian cancer progression would be informative for the readers at first glance.

Response 8: Thank you for pointing this out. We have included Figure 4 in the manuscript (Conclusion section) to summarize the aberrations in the metabolism-related gene expression in OC observed in our study. Changes in the level of genes related with metabolic pathways in ovarian cancer tissue are indicated with arrows: ↑ - increase; ↓ - decrease; ↔ unchanged.

Once again, thank You for all the comments and suggestions on our manuscript. We have tried to make all the issues You have raised in the review. In case of any more corrections, we are open to further cooperation.

Round 2

Reviewer 1 Report

Despite my initial disappointment, the authors have addressed my concerns surrounding the significance nad novelty of the results in the revised version of the manuscript.